# Sexually transmitted and blood-borne infections in transgender and non-binary people in Canada: A scoping review

Jacob Bigio[1]*, Megan Butler[1], Swati Sood[2], Joseph Cox[3], Beth Jackson[4], Zack Marshall[5,6], Josephine Aho[1]

1 Sexually Transmitted and Blood-Borne Infections Surveillance Division, Public Health Agency of Canada, Montreal, Québec, Canada, 2 Library Services Division, Health Canada, Ottawa, Ontario, Canada, 3 Department of Epidemiology, Biostatistics & Occupational Health, McGill University, Montreal, Québec, Canada, 4 Health Equity Policy Directorate, Public Health Agency of Canada, Ottawa, Ontario, Canada, 5 Department of Community Health Services, University of Calgary, Calgary, Alberta, Canada, 6 School of Social Work, McGill University, Montreal, Québec, Canada

* jacob.bigio@proton.me

## Abstract

In the 2021 census, 100,815 people in Canada aged 15 and older identified as transgender or non-binary. Globally, transgender people are disproportionately burdened by several Sexually Transmitted and Blood-Borne Infections (STBBIs) but limited data are available in Canada. Transgender and non-binary people are recognised as key populations in the STBBI Action Plan 2024–2030. We conducted a scoping review of the evidence relating to STBBI prevalence; risk exposures; and use of STBBI testing, treatment and prevention services among transgender and non-binary people in Canada. We searched six databases for articles published between January 1, 2013 and September 1, 2023 and conducted a grey literature search of information published on provincial and territorial public health department websites and websites of eight relevant community organisations. 26 of 934 screened records were included, of which five were provincial surveillance reports from Ontario and Quebec. It is difficult to quantify the prevalence of any STBBI among transgender and non-binary people in Canada. Most provinces and territories, and the federal government, do not publish disaggregated STBBI prevalence data for these populations. Peer-reviewed literature provides HIV prevalence data for several subgroups of the transgender population in some parts of Canada but, in general, these studies were not designed to produce valid prevalence estimates. Transgender people may be less sexually active than other population groups, though this may vary between subgroups of the transgender population. Transgender people face many barriers to accessing healthcare, testing and treatment for STBBIs, while almost no research has been conducted on STBBIs in non-binary people. It is not possible to make specific recommendations for public policy based on the evidence as it currently exists. More public health surveillance

**Data availability statement:** All relevant data are within the manuscript and its Supporting Information files.

**Funding:** The author(s) received no specific funding for this work.

**Competing interests:** The authors have declared that no competing interests exist.

and research, conducted in collaboration with transgender and non-binary communities across Canada, would help to better understand the burden of STBBIs in these populations nationally.

## Introduction

Canada was the first country in the world to provide census data on transgender and non-binary people [1]. In the 2021 census, 100,815 people in Canada aged 15 and older identified as transgender or non-binary, accounting for 0.33% of the population [1]. One of the priorities of the Federal 2SLGBQTI+ Action Plan, released in 2022, is to strengthen data and evidence-based policy making by improving data collection, analysis, research, and knowledge on Two-Spirit, lesbian, gay, bisexual, transgender, queer, intersex and other people who identify as part of sexual and gender diverse (2SLGBQTI+) communities in Canada [2]. In February 2024, the Government of Canada's Sexually Transmitted and Blood-Borne Infections (STBBI) Action Plan for 2024–2030 was released [3], guided by the earlier Pan-Canadian STBBI Framework for Action [4]. The STBBI Action Plan notes that STBBIs do not affect all people equally and that contextual factors, like the social determinants of health, and legal, political and historical factors, influence vulnerability and resilience to STBBIs [3]. The STBBI Action Plan identifies several key populations affected by STBBIs in Canada, including transgender and non-binary people.

On a global scale, the World Health Organization (WHO) describes transgender and gender diverse people as a key population at risk of human immunodeficiency virus (HIV) and other STBBIs [5]. The WHO highlights the role of stigma and discrimination in reducing access to testing, prevention, and treatment services; in creating barriers to safer sex and safer injecting; and in leading to increased risk exposures, including unprotected sex, among transgender and gender diverse people [5]. Systematic reviews of global data show that transgender people are disproportionately burdened by HIV [6], hepatitis B virus (HBV) and hepatitis C virus (HCV) [7] and that data on other STBBIs are limited [8]. Other systematic reviews have shown that transgender people experience high levels of stigma and discrimination in healthcare settings [9] and that stigma increases the vulnerability of transgender women to HIV [10]. However, almost none of the studies included in these systematic reviews were conducted in Canada.

In this scoping review, we explore the evidence published in the past ten years on STBBI prevalence; risk exposures; treatment for STBBIs; testing for STBBIs; and use of healthcare and prevention services, among transgender and non-binary people in Canada.

## Methods

We conducted a scoping review of the literature to identify publications relating to STBBIs in transgender and non-binary people in Canada corresponding to five topic areas:

1. STBBI prevalence

2. Sexual and injection drug use-related risk exposures

3. Testing for STBBIs (including barriers and facilitators to accessing testing)

4. Treatment for STBBIs (including barriers and facilitators to accessing treatment)

5. Use of prevention and other healthcare services (including pre-exposure and post-exposure prophylaxis)

We compiled a list of 21 STBBIs and related conditions: bacterial vaginosis; chlamydia; epididymitis; gonorrhoea; granuloma inguinale; HIV/acquired immunodeficiency syndrome (AIDS); hepatitis A virus (HAV); HBV; HCV; herpes simplex virus (HSV); human papillomavirus (HPV); human T-lymphotropic virus (HTLV); molluscum contagiosum; pelvic inflammatory disease; phthirus; proctitis; scabies; syphilis; trichomoniasis; urethritis; and vaginitis.

We did not publicly register a review protocol. We report our review according to the Preferred Reporting Items for Systematic Reviews and Meta-analyses Extension for Scoping Reviews (PRISMA-ScR) [11].

## Search strategy

In consultation with a librarian (SS), a search strategy was developed to identify relevant literature in six databases (MEDLINE (Ovid); Embase; CAB Abstracts; Global Health; PsycInfo; and SCOPUS) using terms relating to transgender and non-binary people, the 21 STBBIs listed above and Canada. The search was limited to results published between January 1, 2013 and September 1, 2023, with no language filters applied. The search period was chosen to focus on estimates of STBBI prevalence, and data on the other topic areas, that are relatively up-to-date. We also conducted a grey literature search of information published on the 10 provincial and three territorial public health department websites and on the websites of eight relevant community organisations (Action Canada for Sexual Health & Rights; BC Hepatitis C Network; Canadian Aboriginal AIDS Network; CATIE; Community-Based Research Centre; Gay Men's Sexual Health Alliance; Ontario HIV Treatment Network; and REACH Nexus) which are all prominent organisations we identified through our searching. We searched each public health and community organisation website using the advanced search function on google.ca. Each web address (e.g., www.albertahealthservices.ca or www.catie.ca) was entered using the "site:" function and was searched using terms relating to transgender people and the 21 STBBIs. Up to 60 results for each search were assessed, as this corresponded to the first page of results on the advanced search function on google.ca. Full search terms and details of the grey literature search are given in (S1 File).

## Study selection

Search results were imported into Covidence (Covidence, Melbourne, Australia) for screening, which automatically de-duplicates results. Two reviewers (JB and MB) then independently conducted title/abstract and full-text screening. One reviewer (MB) conducted the grey literature search, flagging potentially relevant webpages which were then reviewed by the second reviewer (JB). A pre-defined set of inclusion and exclusion criteria were created to guide the assessment of each article or webpage, though the criteria were refined during the screening process in line with standard practice for scoping reviews. Conflicts at each stage were resolved through discussion between the two reviewers. One reviewer (MB) also screened the reference lists of all included studies for relevant papers and screened reference lists and other data sources mentioned in the grey literature for relevant information. Potentially appropriate sources were then reviewed by the second reviewer (JB).

We did not prespecify our own definitions of "transgender" and "non-binary" but used the terms as they were presented in the literature. We included qualitative, quantitative and mixed-methods studies relating to the five topic areas mentioned above, including theses and dissertations. We excluded articles or webpages that were published before 2013; were not about transgender or non-binary people; did not contain disaggregated data on transgender or non-binary people; did not

contain disaggregated data on Canada; were editorials or commentaries with no quantitative or qualitative data; did not include outcomes relating to the five topic areas mentioned above; or were case studies or short case series (with fewer than ten participants). We also excluded literature reviews and other forms of evidence synthesis identified through title/abstract screening after screening their reference lists for relevant studies. Conference or poster abstracts for which a full article from the same study was already contained in our search were excluded as duplicates. If the full article was not located in our search, we attempted to find one or locate the full poster and contacted authors for further details. When further details could not be retrieved, we excluded the abstract.

### Data extraction

Data were extracted using a standardised form created in Microsoft Word (Microsoft Corporation, Redmond, WA, USA). Data extraction for each included study was performed by one of the two reviewers (JB and MB) and checked by the other reviewer. Extracted data included study location; study name; study time period; study design; population characteristics; STBBI-related findings; and notes on the definition of transgender and non-binary people or other relevant terms used in the paper. For some grey literature results from provinces and territories, a custom extraction form was created corresponding to the nature of the available data.

### Data synthesis

Extracted data were summarised in descriptive tables. Due to substantial heterogeneity between study settings and outcome measures, no quantitative data syntheses were undertaken. Included studies used different terminology when referring to populations of men who have sex with men. For consistency, we have used the term gay, bisexual and other men who have sex with men (GBMSM) throughout this manuscript. For other populations, we used terminology as presented in the studies.

### Expert interviews

Alongside our scoping review, we also interviewed a convenience sample of 18 experts on how STBBIs affect transgender and non-binary people in Canada. The aim of the interviews was to understand their views on the state of the field and to learn about recent and ongoing research on the topic. Twelve (67%) of the experts were academic researchers, five (28%) worked with transgender-focused community organisations and one (6%) worked in government. Around half of the experts identified as transgender or non-binary.

Interviews were semi-structured. We asked the experts for their views on the state of STBBI research and surveillance among transgender and non-binary people in Canada; about data gaps in the field; about recent and ongoing research in the field; and for suggestions for future work they thought would be valuable. We prepared an interview guide that asked broadly about these topics but discussions were largely guided by the responses of the experts towards the topics they felt were most important in the field. We chose this interview strategy because it mirrored the search strategy of our scoping review, which was deliberately broad and exploratory, aiming to identify data on a range of factors affecting STBBI epidemiology and care in these populations without pre-specifying precise outcomes we were searching for. Interviews were conducted by two authors (JB and MB), in some cases in conjunction with a third author (JA), between December 2023 and April 2024. Interviews lasted between 30 and 60 minutes and were conducted virtually.

## Results

After deduplication of search results from databases, 310 unique records were identified for screening. Of these, 248 records were excluded after title/abstract screening, leaving 62 records (of which 13 were conference abstracts). Sixteen of the 62 records met our eligibility criteria for inclusion. Of the 46 excluded records, the main reasons for exclusion were duplicate (n = 14), a lack of data on our topics of interest (n = 12), a lack of disaggregated data on Canada (n = 5) and a

lack of disaggregated data on transgender or non-binary people (n = 5). The full list of reasons for exclusion is given in the PRISMA flowchart shown in Fig 1.

In addition, 259 records were identified through the grey literature searches, citation searching and through requesting further details from the authors of conference abstracts. Of these, five studies and five provincial reports met our eligibility criteria for inclusion, resulting in a total of 26 included publications. The 21 studies and three of the provincial reports were published in English, while two of the provincial reports were published in French. Full details on reasons for exclusion for the studies identified via databases are given in (S2 Table). Reasons for exclusion were not recorded for studies identified via other methods.

## Main characteristics of included publications

Of the 26 included publications, 21 were studies and five were provincial surveillance reports, three from the province of Ontario [12–14] (Table 1 ) and two from the province of Quebec [15,16] (Table 2). The province of Ontario reported on gonorrhoea, syphilis and chlamydia [12–14]. The province of Quebec reported on HIV [15,16].

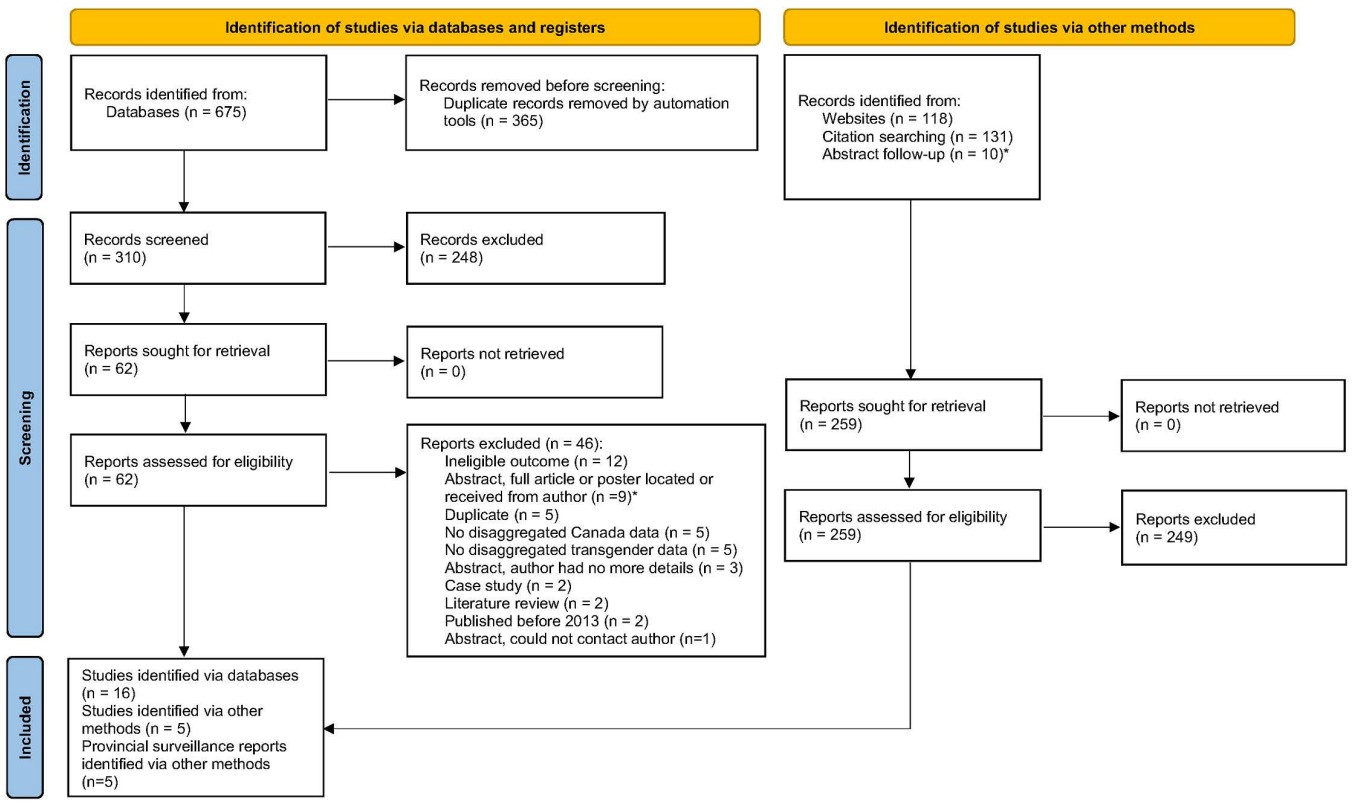

**Fig 1. PRISMA flowchart.** Note that data from one abstract was published as two separate papers.

**Table 1. Number of STBBI cases recorded among the transgender population in Ontario, 2019-2021.**

|  | 2019 | 2020 | 2021 |
| --- | --- | --- | --- |
| Gonorrhoea [12] | 39 | 32 | 36 |
| Syphilis [13] | 13 | 12 | 18 |
| Chlamydia [14] | 32 | 33 | 46 |

**Table 2. Distribution of cases of HIV among transgender people by year in Quebec.**

| Year | New Diagnoses | Past Diagnoses |
|---|---|---|
| 2002* | 0 | 0 |
| 2003 | 0 | 1 |
| 2004 | 0 | 0 |
| 2005 | 0 | 0 |
| 2006 | 0 | 1 |
| 2007 | 0 | 0 |
| 2008 | 0 | 2 |
| 2009 | 0 | 0 |
| 2010 | 0 | 0 |
| 2011 | 0 | 0 |
| 2012 | 0 | 0 |
| 2013 | 0 | 0 |
| 2014 | 0 | 0 |
| 2015 | 0 | 2 |
| 2016 | 0 | 2 |
| 2017 | 0 | 0 |
| 2018 | 1 | 2 |
| 2019 | 3 | 2 |
| 2020 | 0 | 0 |
| 2021 | 3 | 3 |
| TOTAL | 7 | 15 |

*began in April 2002.

Separate data source for 2002–20 [15] and for 2021 [16].

Of the 21 included studies, five described results from the Trans PULSE Ontario study (which was conducted in 2009–10, but met our inclusion criteria as many of the resulting papers were published in 2013 or later) [17–21] (Table 3), three described results from the Montreal Toronto Trans Study (conducted in 2018–19) [22–24] (Table 4), two described results from Sex Now surveys conducted in 2015 [25] and 2018 [26] (Table 5) and 11 described results from other studies [27–37] (Table 6).

Of the studies that included transgender people, six (29%) included the overall transgender population, six (29%) included only transgender men who identified as GBMSM and nine (43%) included only transgender women (without limiting participation to those with specific sexual orientation or activities). Three publications (14%) included non-binary people. Nineteen (90%) of the studies had data collection in Ontario, 10 (48%) had data collection in British Columbia (BC), 11 (52%) had data collection in Quebec and six (29%) had data collection in other provinces and territories. 15 (71%) of the studies were cross-sectional studies, three (14%) were retrospective chart reviews, two (10%) were cross-sectional studies nested within cohort studies and one (5%) was a cohort study.

Nineteen (90%) of the studies reported on HIV or outcomes in people living with HIV (PLHIV), of which 12 (57%) reported only on HIV and no other STBBIs. Four studies (19%) reported on HCV [24,26,29,30], two (10%) reported on HBV [24,26], one (5%) reported on gonorrhoea, chlamydia, syphilis, HBV and HPV [26], one (5%) reported on Mpox [31] and one (5%) reported on sexual health service or primary healthcare use among transgender people in general without specifying any particular STBBI [37]. Fourteen (67%) of the studies collected only quantitative data, six (29%) collected only qualitative data and one (5%) employed a mixed-methods approach. The population type, STBBIs examined, and study location of included studies is shown in Fig 2.

**Table 3. Main features and findings of publications reporting results from the Trans PULSE Ontario study.**

| Paper | Study design, location, study name and study period | Population Characteristics | STBBI-related findings (95% CI) | Notes |
|---|---|---|---|---|
| Longman Marcellin 2013 [17] | Cross-sectional study (Trans PULSE Ontario) from May 2009–April 2010 of 433 transgender people who lived, worked or received healthcare in Ontario. RDS with 38 initial participants who each recruited up to 3 new participants, resulting in 10 waves. | Transgender people, n = 433 Female-to-male: 53% Male-to-female: 47% Aboriginal: 7% | HIV-related sexual risk in the past year: No risk (no sex): 37% (29–46%) Low risk: 51% (42–59%) High risk: 12% (7–18%) | Transgender people defined as those "who identified as trans or as having trans experience." Transgender GBMSM defined as those "who indicated they had a sexual minority identity (e.g., gay, bisexual, pansexual, queer) and were not exclusively attracted to cis women, or those who had had sex with a cis or trans man in the past year, regardless of how they identified." High risk sexual activity defined "according to Canadian AIDS Society (2004) guidelines as unprotected (fluid exposed) sex outside of a HIV-seroconcordant monogamous relationship." Aboriginal people were defined as people who "indicated being either a member of a Canadian Aboriginal group (First Nations, Métis, or Inuit), or of other Indigenous ancestry." |
| Scheim 2017a [18] | Sub-analysis of Trans PULSE Ontario data relating to transgender GBMSM | Transgender GBMSM, n = 158 Mean age: 29.8 years Aboriginal: 3.7% | HIV-related sexual risk in the past year: No risk (no sex): 15.0% (6.3–23.7%) Low risk: 75.0% (64.2–85.7%) High risk: 10.0% (1.5–18.6%) HIV testing: Past year: 18.7% (9.2–28.3%) >1 year ago: 41.9% (29.0–54.8%) Never: 39.3% (26.4–52.2%) Self-reported HIV status: Positive: 0.0 Negative: 82.8% (72.9–92.6%) Don't know or prefer not to say: 17.2% (7.4–27.1%) | |
| Scheim 2017b [19] | Sub-analysis of Trans PULSE Ontario data relating to transfeminine people | Transgender women, n = 171 Mean age: 39 years Indigenous: 10% | Self-reported HIV status: Negative: 75.2% (64.2–86.2%) Positive: 1.2% (0.0–3.0%) Did not know their status: 23.6% (12.7–34.6%) | |
| Scheim 2018 [20] | Sub-analysis of Trans PULSE Ontario data relating to transgender women | Transgender women, n = 171 Mean age: 39 years Indigenous: 10% | HIV-related sexual risk: No sex (sexually inactive): 41% (29–53%) Low risk: 38% (27–50%) High risk: 21% (12–30%) | |
| Munro 2017 [21] | Qualitative phase of Trans PULSE Ontario involving semi-structured interviews with 14 trans WLWH on barriers to healthcare utilization and 10 service providers who work with trans WLWH | Trans WLWH, n = 14 Age 30–39: 43%; 40–49: 50%; 50–59: 7% Aboriginal, First Nations, Native or Metis 29% | "Four themes [were identified] related to challenges in interactions with health care providers and other workers in community health center or community agency settings: (a) discrimination in health care encounters, (b) denial of service, (c) lack of training impacts HIV clinicians and trans people, and (d) agencies not perceived as welcoming to trans women." "Five themes [were identified] related to systemic barriers in navigation of health systems: (a) lack of trans-specific services, (b) disjointed health care delivery models, (c) siloing reinforces limited service options, (d) system navigation complicated by discrimination, and (e) geographic barriers to HIV testing and care." | The term trans is used to "refer to a variety of people who do not identify with the gender they were assigned at birth. This definition of trans is inclusive of people who express as gender ambiguous, genderqueer, or Two-Spirit, and does not imply that individuals need to be involved in surgical or hormonal transition." |

AIDS = acquired immunodeficiency syndrome; CI = confidence interval; cis = cisgender; GBMSM = gay, bisexual and other men who have sex with men; HIV = human immunodeficiency virus; RDS = respondent-driven sampling; STBBI = sexually transmitted and blood-borne infection; WLWH = women living with HIV.

**Table 4. Main features and findings of publications reporting results from the Montreal Toronto Trans Study.**

| Paper | Study design, location, study name and study period | Population Characteristics | STBBI-related findings (95% CI) | Notes |
|---|---|---|---|---|
| Armstrong 2023 [22] | Retrospective chart review (Montreal Toronto Trans Study) of data on 1495 transgender women aged ≥16 years from seven family medicine, endocrinology or HIV clinics in Montreal and Toronto from July 2018 – December 2019 | Transgender women, n = 1495 Median age (IQR): 35 (27, 47) | HIV status: Positive HIV test on file: 5.8%; Negative HIV test on file: 40.1%; No documented HIV test on file: 54.2% | No definition of transgender women was provided. The list of trans women was created by the clinical sites "on the basis of their standard EMR documentation procedures and confirmation from relevant care providers" Duplicate patients (those receiving care at multiple sites) were identified and merged using clinic and linking logs. The chosen clinics "represented the primary clinical care sites for the Majority of TWLWH in Montreal and Toronto" |
| Lacombe-Duncan 2022 [23] | Sub-analysis of Montreal Toronto Trans Study data relating to transgender WLWH | Transgender WLWH, n = 86 Age < 30: 19.8%; 30–39: 23.3%; 40–49: 25.6%; 50 +: 31.4% | Ever engaged in HIV care: 86/86 (100%) Currently using ART: 80/86 (93.0%) Suppressed viral load: 82.6% Unsuppressed viral load: 5.8% Missing: 11.6% History of injection drug use: Yes: 8.1%; No: 37.2%; Missing: 54.7% | |
| Lacombe-Duncan 2023 [24] | As Armstrong 2023 but excluded participants who only saw an endocrinologist due to the amount of missing HIV-related data at this site | Transgender women, n = 1059 Age < 30: 32.5%; 30–50: 47.4%; > 50: 19.8%; Missing: 0.3% | HIV status: PLHIV: 7.5%; HIV negative 54.4%; Missing 38.1% Past history of HCV: Yes: 2.8%; No: 55.5%; Missing: 41.6% Past history of HBV: Yes: 1.7%; No: 51.6%; Missing: 46.7% History of sex work: Ever: 10.5%; Never 4.2%; Missing 85.3% | |

ART = antiretroviral therapy; CI = confidence interval; EMR = electronic medical records; HBV = hepatitis B virus; HCV = hepatitis C virus; HIV = human immunodeficiency virus; IQR = interquartile range; Missing = missing data; PLHIV = people living with HIV; STBBI = sexually transmitted and blood-borne infection; TWLWH = transgender women living with HIV; WLWH = women living with HIV.

## STBBI prevalence

Three studies provided prevalence estimates for the overall transgender population, rather than subgroups of the transgender population. Abramovich 2020, a cross-sectional study of service use in Ontario, found that the diagnosed prevalence of HIV was much higher in 2,085 transgender people attending four outpatient clinics (1.6%) than in 10,425 age-matched presumed cisgender people in the general provincial population (0.1%) (p-value: < 0.001) [27]. In Scruton 2014, a cross-sectional online survey of transgender people in all provinces and the Northwest Territories, 0.8% of 460 participants self-reported being HIV positive, with 17.9% unaware of their HIV status and 1.9% preferring not to answer [36]. Cardinal 2021, a retrospective study of data from 11 clinical cohorts of PLHIV in five provinces, found that 36–45% of the 56 transgender PLHIV had ever been co-infected with HCV [29].

In surveillance data, the province of Ontario reported 36 cases of gonorrhoea, 18 cases of syphilis and 46 cases of chlamydia among transgender people in 2021; 32, 12 and 33 cases, respectively, in 2020; and 39, 13 and 32 cases, respectively, in 2019 [12–14]. In surveillance data, the province of Quebec reported seven newly diagnosed cases of HIV in transgender people in the period 2002–2021 [15,16].

Three studies provided prevalence estimates for transgender women. Armstrong 2023, a retrospective chart review of 1,495 transgender women attending seven endocrinology or HIV primary care clinics in Montreal and Toronto (the Montreal Toronto Trans Study), found that 5.8% had a positive HIV test on file, 40.0% had a negative HIV test on file and 54.2% did not have results on file [22]. Lacombe-Duncan 2023, which was a sub-analysis of data relating to 1,059 of the

**Table 5. Main features and findings of publications reporting results from Sex Now surveys.**

| Paper | Study design, location, study name and study period | Population Characteristics | STBBI-related findings (95% CI) | | | | Notes |
|---|---|---|---|---|---|---|---|
| Ferlatte 2020 [25] | Cross-sectional national online survey (Sex Now 2015) comparing sexual health outcomes of cisgender and trans-gender GBMSM from November 2014 – April 2015. | Cisgender GBMSM, n = 7439 Age < 30: 23.5%; 30–45: 27.7%; 45 +: 48.8% Indigenous: 4.0% Transgender GBMSM, n = 209 Age < 30: 66.0%; 30–45: 17.2%; 45 +: 16.7% Indigenous: 4.3% | Category Sexual activities in the last 12 months: | Cis | Trans | aOR (95% CI) Trans versus cis: | "Transgender participants were defined as those who responded "transgender" in the gender identity question. Respondents who also selected woman in response to the gender identity question were excluded from the analysis." Odds ratios adjusted for age and ethnicity. |
| | | | Not sexually active: | 6.8% | 20.6% | 3.82 (2.67–5.46) | |
| | | | Casual partners: | 64.8% | 37.3% | 0.34 (0.25–0.45) | |
| | | | Sex with primary partner: | 43.8% | 40.7% | 0.72 (0.54–0.95) | |
| | | | Fuck buddies: | 50.4% | 33.5% | 0.49 (0.37–0.66) | |
| | | | Group sex: | 29.2% | 9.1% | 0.24 (0.15–0.39) | |
| | | | 20 + sex partners: | 16.6% | 3.3% | 0.19 (0.09–0.41) | |
| | | | Intercourse without condoms in the last 12 months: | | | | |
| | | | Same HIV status partner: | 53.6% | 40.7% | 0.50 (0.37–0.66) | |
| | | | Unknown HIV status partner: | 22.3% | 11.5% | 0.43 (0.28–0.66) | |
| | | | Different HIV status partner: | 10.6% | 2.4% | 0.23 (0.10–0.56) | |
| | | | Testing in the last 12 month: | | | | |
| | | | STI Test: | 57.7% | 47.0% | 0.70 (0.53–0.92) | |
| | | | HIV Test: | 59.0% | 47.8% | 0.64 (0.49–0.85) | |
| | | | Diagnosed with STI: | 15.1% | 5.7% | 0.35 (0.20–0.64) | |
| Rutherford 2021 [26] | Cross-sectional survey (Sex Now 2018) of cisgender and trans-gender men, non-binary and Two-Spirit people aged ≥15 years old who "identify as gay, bisexual, queer, or another non-heterosexual identity and/or have reported having had sex with another man (cis or trans) in the last 5 years." In-person recruitment at Pride festivals and related events in 15 Canadian cities from May – September 2018. | Cisgender participants, n = 3083 Age Group <25: 18.0%; 25–29: 20.9%; 30–39: 28.6%; 40–49: 12.3%; 50–59: 12.9%; 60 +: 6.7% Indigenous: 8.1% Trans participants, n = 296 Age Group <25: 42.2%; 25–29: 17.9%; 30–39: 19.9%, 40–49: 8.8%; 50–59: 7.1%; 60 +: 3.0% Indigenous: 16.6% Non-binary participants, n = 150 Age Group <25: 40.7%; 25–29: 25.3%; 30–39: 18.7%, 40–49: 7.3%; 50–59: 3.3%; 60 +: 3.3% Indigenous: 16.0% Participants who identified both as non-binary and as having trans experience, n = 106 | Category Chlamydia, gonorrhea, or syphilis, past year: | Cis | Trans | aOR (95% CI) | Participants were grouped into cisgender, transgender or non-binary categories "based on their responses to two survey questions: gender identity and trans-gender lived experience." Participants who had partic-ipated at one study venue could not participate again at another venue. ORs were adjusted for age, white ethnoracial identity vs. non-white ethnoracial identity, born in Canada vs. not, gay sexual identity vs. not, and financial strain. ORs displayed in bold were statistically significant at the 5% level. Percentages may not add up to 100% due to variable amounts of missing data per question. |
| | | | Yes: | 16.4% | 9.8% | 0.89 (0.57–1.36) | |
| | | | No: | 72.8% | 70.3% | 1.00 | |
| | | | Timing of last STI test: | | | | |
| | | | In the past year: | 68.5% | 59.8% | 1.00 | |
| | | | > than a year ago: | 20.1% | 16.2% | 0.92 (0.63–1.33) | |
| | | | Never: | 7.0% | 15.5% | 1.05 (0.68–1.58) | |
| | | | Ever tested for HCV: | | | | |
| | | | Yes: | 72.7% | 60.5% | 1.00 | |
| | | | No: | 8.5% | 17.9% | 1.20 (0.79–1.77) | |
| | | | Ever diagnosed with HCV: | | | | |
| | | | No: | 71.7% | 55.7% | 1.00 | |
| | | | Yes: | 0.8% | 1.4% | 1.08 (0.28–3.41) | |
| | | | Ever tested for HIV: | | | | |
| | | | Yes: | 87.3% | 70.9% | 1.00 | |
| | | | No: | 7.5% | 16.2% | 0.88 (0.58–1.31) | |
| | | | Ever diagnosed with HIV: | | | | |
| | | | No: | 79.8% | 65.2% | 1.00 | |
| | | | Yes: | 6.5% | 5.1% | 1.43 (0.75–2.56) | |
| | | | Ever used HIV PrEP: | | | | |

*(Continued)*

| Paper | Study design, location, study name and study period | Population Characteristics | STBBI-related findings (95% CI) | | | | Notes |
|---|---|---|---|---|---|---|---|
| | | | No: | 74.6% | 83.1% | 1.00 | |
| | | | Yes, but I stopped: | 2.6% | 4.4% | **2.96 (1.45–5.65)** | |
| | | | Yes, I'm taking PrEP now: | 12.3% | 4.1% | 0.56 (0.28–1.00) | |
| | | | Has a regular HCP: | | | | |
| | | | No: | 24.4% | 19.9% | 1.00 | |
| | | | Yes: | 71.7% | 76.4% | **1.82 (1.30–2.57)** | |
| | | | Ever vaccinated for HBV: | | | | |
| | | | No: | 11.9% | 13.5% | 1.00 | |
| | | | Unsure: | 14.9% | 22.3% | 1.23 (0.77–1.99) | |
| | | | Yes: | 67.9% | 58.4% | 1.15 (0.77–1.76) | |
| | | | Ever vaccinated for HPV: | | | | |
| | | | No: | 39.6% | 29.1% | 1.00 | |
| | | | Unsure: | 19.6% | 25.3% | 1.21 (0.83–1.74) | |
| | | | Yes: | 30.6% | 38.2% | **1.51 (1.08–2.12)** | |
| | | | Asked and Denied: | | | | |
| | | | An HIV test: | 3.3% | 5.7% | 1.29 (0.68–2.30) | |
| | | | PEP: | 1.0% | 2.7% | 2.29 (0.87–5.42) | |
| | | | PrEP: | 2.3% | 2.4% | 0.99 (0.37–2.25) | |
| | | | HPV vaccination: | 1.6% | 4.1% | **2.78 (1.25–5.79)** | |
| | | | Category | Cis | Non-binary | aOR (95% CI) | |
| | | | Chlamydia, gonorrhea, or syphilis, past year: | | | | |
| | | | Yes: | 16.4% | 12.0% | 1.04 (0.58–1.80) | |
| | | | No: | 72.8% | 71.3% | 1.00 | |
| | | | Timing of last STI test: | | | | |
| | | | In the past year: | 68.5% | 62.0% | 1.00 | |
| | | | > than a year ago: | 20.1% | 16.7% | 0.86 (0.50–1.42) | |
| | | | Never: | 7.0% | 13.3% | 0.76 (0.41–1.34) | |
| | | | Ever tested for HCV: | | | | |
| | | | Yes: | 72.7% | 62.7% | 1.00 | |
| | | | No: | 8.5% | 12.0% | 0.75 (0.39–1.36) | |
| | | | Ever diagnosed with HCV: | | | | |
| | | | No: | 71.7% | 58.0% | 1.00 | |
| | | | Yes: | 0.8% | 2.0% | 2.43 (0.46–9.85) | |
| | | | Ever tested for HIV: | | | | |
| | | | Yes: | 87.3% | 72.7% | 1.00 | |
| | | | No: | 7.5% | 15.3% | 0.80 (0.45–1.36) | |
| | | | Ever diagnosed with HIV: | | | | |
| | | | No: | 79.8% | 68.7% | 1.00 | |
| | | | Yes: | 6.5% | 5.3% | 0.68 (0.65–3.81) | |
| | | | Ever used HIV PrEP: | | | | |
| | | | No: | 74.6% | 82.0% | 1.00 | |
| | | | Yes, but I stopped: | 2.6% | 3.3% | 2.34 (0.73–6.06) | |

*(Continued)*

**Table 5.** (Continued)

| Paper | Study design, location, study name and study period | Population Characteristics | STBBI-related findings (95% CI) | | | | Notes |
|---|---|---|---|---|---|---|---|
| | | | Yes, I'm taking PrEP now: | 12.3% | 4.7% | 0.73 (0.29–1.53) | |
| | | | Has a regular HCP: | | | | |
| | | | No: | 24.4% | 26.7% | 1.00 | |
| | | | Yes: | 71.7% | 71.3% | 1.53 (1.00–2.38) | |
| | | | Ever vaccinated for HBV: | | | | |
| | | | No: | 11.9% | 14.0% | 1.00 | |
| | | | Unsure: | 14.9% | 22.7% | 1.18 (0.63–2.24) | |
| | | | Yes: | 67.9% | 58.7% | 1.09 (0.64–1.94) | |
| | | | Ever vaccinated for HPV: | | | | |
| | | | No: | 39.6% | 27.3% | 1.00 | |
| | | | Unsure: | 19.6% | 25.3% | 1.29 (0.77–2.14) | |
| | | | Yes: | 30.6% | 36.0% | 1.47 (0.92–2.36) | |
| | | | Asked and Denied: | | | | |
| | | | An HIV test: | 3.3% | 6.0% | 1.40 (0.60–2.94) | |
| | | | PEP: | 1.0% | 2.0% | 1.69 (0.35–5.77) | |
| | | | PrEP: | 2.3% | 2.0% | 0.87 (0.19–2.67) | |
| | | | HPV vaccination: | 1.6% | 3.3% | 1.87 (0.56–5.19) | |

aOR = adjusted odds ratio; CI = confidence interval; cis = cisgender; GBMSM = gay, bisexual and other men who have sex with men; HBV = hepatitis B virus; HCP = healthcare provider; HCV = hepatitis C virus; HIV = human immunodeficiency virus; HPV = human papillomavirus; OR = odds ratio; PEP = post-exposure prophylaxis; PrEP = pre-exposure prophylaxis; STBBI = sexually transmitted and blood-borne infection; STI = sexually transmitted infection.

same transgender women, found that 1.7% had a past history of HBV and 2.8% had a past history of HCV, though data were missing for 46.7% and 41.6%, respectively [24]. Scheim 2017b, which was a sub-analysis of data from the Trans PULSE Ontario study focusing on 171 transgender women, found that 1.2% (95% confidence interval (CI) 0.0–3.0%) of participants self-reported as living with HIV while 23.6% (12.7–34.6%) did not know their status or preferred not to answer [19].

Three studies provided prevalence estimates for transgender GBMSM, one of which also provided estimates for non-binary people who identified as GBMSM or as having had sex with another man in the past five years. National data from the in-person Sex Now 2018 survey published by Rutherford 2021 found that 5.1% of 296 transgender GBMSM self-reported having ever been diagnosed with HIV, 1.4% had ever been diagnosed with HCV and 9.8% had been diagnosed with chlamydia, gonorrhoea or syphilis in the past year. The same survey found that 5.3% of 150 non-binary GBMSM had ever been diagnosed with HIV, 2.0% had ever been diagnosed with HCV and 12.0% had been diagnosed with chlamydia, gonorrhoea or syphilis in the past year. These figures for transgender and non-binary GBMSM were not statistically different from those of cisgender GBMSM in the same survey. One hundred and six participants identified as both transgender and non-binary, so the findings of those groups overlapped [26]. National data from the online Sex Now 2015 survey published by Ferlatte 2020 found that 5.7% of 209 transgender GBMSM and 15.1% of 743 cisgender GBMSM self-reported having been diagnosed with a sexually transmitted infection (STI) in the past year, giving an adjusted odds ratio (aOR) of 0.35 (0.20–0.64), though statistically significantly fewer transgender GBMSM had tested in the past year [25]. Scheim 2017a, which was a sub-analysis of data from the Trans PULSE

Ontario study focusing on 158 transgender GBMSM, found that no participants self-reported as being HIV positive while 17.2% (7.4–27.1%) did not know their status or preferred not to answer [18].

## Sexual and injection drug use-related risk exposures

In the Trans PULSE Ontario study, HIV-related sexual risk in the past year was categorised as high, low or no risk. High HIV-related sexual risk was defined as "unprotected (fluid exposed) sex outside of a HIV-seroconcordant monogamous relationship", low risk was having had sex but not according to the high risk conditions and no risk was having not had sex in the past year. Thirty-seven percent (29–46%) of 433 transgender people surveyed had no HIV-related sexual risk, while 12% (7–18%) had high HIV-related sexual risk in the past year [17]. However, there was variation between subgroups of the transgender population, which were examined in separate papers: Scheim 2017a found that only 15.0% (6.3–23.7%) of transgender GBMSM in the study had no HIV-related sexual risk while 10.0% (1.5–18.6%) had high HIV-related sexual risk [18]. By contrast, Scheim 2018 found that 41% (29–53%) of transgender women in the study had no HIV-related sexual risk and 21% (12–30%) had high HIV-related sexual risk [20]. In Scruton 2014, 10% of 460 transgender people had had unprotected sex with a high-risk partner or a partner who does not know their HIV status in the past 12 months. No definition of the term "high-risk partner" was given. Thirty-four percent had had no sex in the past 12 months [36]. In Arps 2021, a sub-analysis of data on 2,369 transgender and non-binary participants aged ≥16 years from the Trans PULSE Canada study, which recruited transgender and non-binary people online and in-person in six provinces, 6% reported having engaged in sex work in the past year [28].

The Sex Now 2015 survey, published by Ferlatte 2020, asked about sexual activity in the past year. Transgender GBMSM had higher odds of not being sexually active than cisgender GBMSM (aOR 3.82 [2.67–5.46]) and lower odds than cisgender GBMSM of having had casual sex partners (aOR 0.34 [0.25–0.45]), fuck buddies (aOR 0.49 [0.37–0.66]), group sex (aOR 0.24 [0.15–0.39]) and 20+sex partners (aOR 0.19 [0.09–0.41]). Transgender GBMSM also had lower odds than cisgender GBMSM of having had intercourse without condoms in the past year with a partner with the same HIV status (aOR 0.50 [0.37–0.66]), with a partner with unknown HIV status (aOR 0.43 [0.28–0.66]) and with a partner with different HIV status (aOR 0.23 [0.10–0.56]) [25].

In Lacombe-Duncan 2023, the sub-analysis of the Montreal Toronto Trans Study, 10.5% of 1,059 transgender women reported ever engaging in sex work, with missing data for 85.3% of participants [24].

Of the 460 transgender people in Scruton 2014, 0.4% reported sharing needles in the past 12 months [36]. In Lacombe-Duncan 2022, a sub-analysis of the Montreal Toronto Trans Study that focused on 86 transgender women living with HIV (WLWH), 8.1% of participants reported a history of injection drug use, with missing data for 54.7% [16].

## Testing for STBBIs

Arps 2021 compared outcomes between transgender and non-binary people who engaged in sex work in the past year, and those who did not. Of the 2,236 transgender and non-binary people in the survey who were not current sex workers, 21% had tested for HIV in the past year and 51% had never had an HIV test. Of the 133 current sex workers in the survey, 50% had tested for HIV in the past year and 29% had never had an HIV test. For non-HIV STI tests, 27% of non-sex workers had tested in the past year and 40% had never tested, while 73% of sex workers had tested in the past year and 13% had never tested [28]. In Scruton 2014, of the 262 transgender people for whom data were available, 25% had tested for HIV within the past year and 39% had never tested for HIV [36].

Scheim 2017a found that 18.7% (9.2–28.3%) of transgender GBMSM had been tested for HIV in the past year and 39.3% (26.4–52.2%) had never been tested for HIV [18]. In the Sex Now 2015 survey, published by Ferlatte 2020, transgender GBMSM had lower odds than cisgender GBMSM of having been tested for HIV (aOR 0.64 [0.49–0.85]) and other STIs (aOR 0.70 [0.53–0.92]) in the past year [25]. In the Sex Now 2018 survey, published by Rutherford 2021, 59.8%

of transgender GBMSM had tested for STIs in the past year and 15.5% had never tested for STIs. Of the non-binary GBMSM in the survey, 62.0% had tested for STIs in the past year and 13.3% had never tested for STIs. These figures for transgender and non-binary GBMSM were not statistically different from those of cisgender GBMSM in the same survey. 60.5% of transgender GBMSM and 62.7% of non-binary GBMSM had ever tested for HCV and 70.9% of transgender GBMSM and 72.7% of non-binary GBMSM had ever tested for HIV and again these figures were not statistically different from those of cisgender GBMSM in the same survey [26].

## Treatment for STBBIs

Cardinal 2021 found that, of the 56 transgender people in the study (who were grouped together separately from the categories of women and men), 58.9% had a suppressed viral load within a year of starting antiretroviral therapy (ART) and no viral load rebound within a year of suppression. This compared with 66.0% of the 1,771 cisgender women in the study and 80.4% of the 8,728 cisgender men in the study, though no measures of association were given comparing the groups [29].

Lacombe-Duncan 2019, a cross-sectional study of 50 transgender WLWH nested within a larger cohort study of cisgender and transgender WLWH in BC, Ontario and Quebec, presented data on the HIV care cascade. 92% of the transgender WLWH had ever accessed HIV care, 78% of those who had accessed care were currently using ART, 67% of those using ART had ≥ 95% ART adherence and 89% of those using ART were virally suppressed [32]. In Lacombe-Duncan 2022, the sub-analysis of the Montreal Toronto Trans Study focusing on 86 transgender WLWH, 93.0% of participants were currently using ART, of whom 82.6% had a suppressed viral load, 5.8% had an unsuppressed viral load and data were missing for 11.6% [23].

## Barriers to accessing testing and treatment for STBBIs

Two studies found that discrimination and stigma experienced in healthcare encounters were barriers to transgender women accessing testing and treatment for STBBIs: Lacombe-Duncan 2021, a qualitative study of 26 transgender women in Vancouver, Toronto and Montreal [33], and Munro 2017, a qualitative sub-study of the Trans PULSE Ontario study involving semi-structured interviews with 14 transgender WLWH on barriers to healthcare utilization and 10 service providers who work with trans WLWH [21].

Participants in four studies focusing on different subpopulations noted that service providers often lacked knowledge about the specific sexual health needs of transgender and non-binary people: Lacombe-Duncan 2021 [33]; Rich 2017, a qualitative sub-study about sex-related HIV risk among 11 transgender GBMSM recruited from the longitudinal Momentum Health Study cohort in Metro Vancouver [34]; Scheim 2017c, a qualitative survey of 40 transgender GBMSM in Ontario [35]; and Stewart 2022, a sub-analysis of qualitative data relating to 13 transgender, non-binary or Two-Spirit GBMSM from a larger cross-sectional qualitative study of STI testing experiences of cisgender and transgender GBMSM in Ontario [34].

In Stewart 2022, participants also described exclusionary policies such as gender segregated clinic hours as barriers to accessing STBBI testing as they "relied on cisnormative expectations that clients identified with one gender, or that clients would automatically know which hours to attend if they were trans" [37].

In Lacombe-Duncan 2019, the study of transgender WLWH, the most commonly cited barrier to ART use was concerns about interactions between ART and feminising hormones [32].

## Facilitators to accessing testing and treatment for STBBIs

Two studies with transgender women noted that the provision of trauma-informed transgender-specific and transgender-inclusive services could act as facilitators to accessing testing and treatment for STBBIs: Everhart 2022,

**Table 6. Main features and findings of other studies reporting on STBBIs in transgender and non-binary people in Canada.**

| Paper | Study design, location, study name and study period | Population Characteristics | STBBI-related findings (95% CI) | | | | Notes |
|---|---|---|---|---|---|---|---|
| Abramovich 2020 [27] | Cross-sectional study comparing service use of transgender people with the general population of Ontario from January 2012–December 2016. Transgender people eligible for Ontario health insurance were identified in 4 outpatient clinics in Ottawa, Thunder Bay and Toronto. Their data were linked with province-wide health administrative data. Each transgender person was age-matched 1:5 to a random 5% sample of the general Ontario population (some of whom may be transgender because the administrative data do not include gender identity). | Transgender people, n = 2,085 Mean age = 30.25 years Sex listed on health card: Female 50.6%, male 49.4% Presumed cisgender people, n = 10,425 Mean age = 30.25 years Sex listed on health card: Female 49.9%, male 50.1% | HIV prevalence in transgender people: 1.6% HIV prevalence in presumed cisgender people: 0.1% SMD: 0.16; p-value: < 0.001 Mean primary care provider visits by transgender people over the study period: 22.55 (SD 27.15) Mean primary care provider visits by presumed cisgender people over the study period: 13.18 (SD 19.62) SMD: 1.91; p-value: 0.40 | | | | "All 4 clinics regularly collect data on self-defined gender identity, making it possible for them to identify their transgender patients within their clinic health records." |
| Arps 2021 [28] | Sub-analysis of data relating to participants aged ≥16 years from a cross-sectional study (Trans PULSE Canada) of 2,873 trans and non-binary people aged ≥14 years who live in Canada. Recruitment was online, in person at sexual and gender minority–focused community spaces and events and through outreach by peer research associates in Vancouver, Calgary, Edmonton, Saskatoon, Winnipeg, Southwestern Ontario, the Greater Toronto Area, downtown Toronto, Ottawa and Montreal, in English and French, from July 26 to October 1, 2019. | Trans and non-binary current sex workers, n=133 Age 14–19: 5%; 20–24: 30%; 25–34: 44%; 35–49: 17%; 50–64: 4%; 65+: 0.8% Indigenous: 7% Trans and non-binary non-sex workers, n=2236 Age 14–19: 10%; 20–24: 22%; 25–34: 37%; 35–49: 21%; 50–64: 8%; 65+: 1% Indigenous: 8% | **Category** | **Sex Workers** | **Non-Sex Workers** | **p-value** | "Participants were able to complete the full survey or a 10-minute short form online, on paper, via telephone (with or without a language interpreter), or on a tablet with a Peer Research Associate (only in major cities)." "Questions on employment and income, including sex work, were limited to participants aged 16 and older." "Sex work was not defined for participants, though those who reported doing sex work were asked to indicate the types of work they were doing." |
| | | | Timing of most recent HIV test: | | | <0.0001 | |
| | | | Less than 1 year ago: | 50% | 21% | | |
| | | | 1 -<2 years ago: | 10% | 9% | | |
| | | | 2+years ago: | 11% | 19% | | |
| | | | Never had an HIV test: | 29% | 51% | | |
| | | | Timing of most recent STI test: | | | <0.0001 | |
| | | | Less than 1 year ago: | 73% | 27% | | |
| | | | 1 -<2 years ago: | 7% | 12% | | |
| | | | 2+years ago: | 6% | 21% | | |
| | | | Never had an STI test: | 13% | 40% | | |
| | | | Ever taken PrEP for HIV prevention: | | | <0.0001 | |
| | | | Yes, currently: | 6% | 0.8% | | |
| | | | Yes, in the past: | 8% | 1% | | |
| | | | No: | 86% | 98% | | |
| | | | Has a primary health care provider: | | | 0.038 | |
| | | | Yes: | 74% | 81% | | |
| | | | No: | 26% | 19% | | |

*(Continued)*

| Paper | Study design, location, study name and study period | Population Characteristics | STBBI-related findings (95% CI) | | | | Notes |
|---|---|---|---|---|---|---|---|
| Cardinal 2021 [29] | Retrospective observational cohort study of data from the Canadian HIV Observational Cohort (CANOC) study, a collaboration of 11 clinical cohorts from British Columbia, Saskatchewan, Ontario, Quebec and Newfoundland and Labrador. CANOC consists of "treatment naïve PLHIV initiating cART between January 1, 2000 and December 31, 2016". Participants with 18+ months of follow up time and a known gender (self-reported) were included. | Men, n=8728 Women, n=1771 Transgender people, n=56 [Demographic information not provided] | Category | Men | Women | Transgender | "Eligible participants had to have at least one plasma viral load result within 1 year of initiating cART and at least 2 consecutive VL results post-initiation at least 30 days apart to assess suppression. Finally, participants who achieved suppression had to have at least one VL result within one year of suppressing and at least 2 consecutive VL results post-suppression at least 30 days apart, to assess rebound within one year of suppression." It is unclear how each clinic recorded gender. "Gender may be self-reported or may default to sex if no gender is indicated." Transgender results in bold were given as ranges to protect participant privacy due to small sample sizes. |
| | | | Ever co-infected with Hepatitis C: | | | | |
| | | | Yes: | 21.3% | 39.7% | **36-45%** | |
| | | | No: | 75.6% | 54.0% | 58.9% | |
| | | | Unknown: | 3.1% | 6.3% | **<9%** | |
| | | | Suppression and rebound: | | | | |
| | | | VL supressed within 12 months and no rebound within 12 months after it: | 80.4% | 66.0% | 58.9% | |
| | | | VL rebound within 12 months after suppressing within 12 months: | 2.6% | 6.8% | 7.1% | |
| | | | Did not achieve VL suppression within 12 months: | 17.0% | 27.2% | **27-36%** | |
| Everhart 2022 [30] | Analysis of qualitative data from a cross-sectional survey of transgender women in Edmonton, Montreal, Toronto, Vancouver and Winnipeg. Unclear timing of interviews | Transgender women, n=76 Age 18–24: 6.8%; 25–34: 24.3%; 35–44: 32.4%; 45–54: 29.7%; 55–64: 5.4%; 65+: 1.4%; No answer: 2.8% Aboriginal or Indigenous: 38.2% | "What was the result of your last HIV test?" Positive: 19.7% Negative: 63.2% Don't want to say: 3.9% Didn't go back: 2.6% Never tested: 6.6% No response: 3.9% "Has a doctor ever told you that you have Hepatitis C?" Yes: 22.4% No: 75.0% No Response: 2.6% "Have you exchanged sex for money or other goods?" Yes: 77.6% No: 18.4% No Response: 4.0% "Key themes that emerged in these discussions highlighted the need for 1) trans-friendly and trans-specific services; 2) integrated health services and aid in navigating complex systems, and; 3) comprehensive, community-based services." | | | | No definition of the term trans was provided. |

*(Continued)*

| Paper | Study design, location, study name and study period | Population Characteristics | STBBI-related findings (95% CI) | Notes |
|-------|------|------|------|------|
| Gilbert 2023 [31] | Cross-sectional online survey from August 8–22, 2022 of Mpox vaccine uptake among trans and GBMSM clients of an STI clinic in Vancouver aged ≥ 18 years who also met one of several eligibility criteria relating to recent bacterial STI diagnosis or sexual risk behaviours | Transgender people, n = 29 Cisgender people, n = 249 Other gender identify/prefer not to say, n = 18 [Disaggregated demographic information not provided] | Received Mpox vaccine among those eligible: Transgender: 10/19 (52.6%) Cisgender: 120/174 (69.0%) Other gender identity/prefer not to say: 7/13 (53.8%) p = 0.212 | No definition of the term transgender was provided. |
| Lacombe-Duncan 2019 [32] | Sub-analysis of cross-sectional data relating to transgender WLWH from the Canadian HIV Women's Sexual and Reproductive Health Cohort Study (CHIWOS) of 1422 cisgender and transgender WLWH aged ≥16 years in Ontario, British Columbia and Quebec Also qualitative semi-structured interview data from a subsample of 11 participants. Quantitative baseline survey data: 2013–2015. Interviews: 2017–2018 | Transgender WLWH, n = 50 Mean age: 41.0 years Indigenous: 34.0% | HIV care cascade: Ever accessed HIV care: 46/50 (92%) Received any HIV care visit within the past year: 42/46 (91%) Currently use ART: 36/46 (78%) ≥95% ART adherence: 24/36 (67%) Virally suppressed 31/35 (89%) [Missing viral load data for one participant]. Qualitative findings (n=11): "The most commonly cited barrier to ART initiation/use among trans WLWH was concern about drug–drug interactions between ART and feminizing hormones" "Participants faced multiple life circumstances that impeded their engagement in HIV care, including trauma, substance use, social isolation, violence, and housing insecurity." "Participants also described both HIV and transphobia in health care as barriers to access to HIV care" "Participants' relationships with HIV health care providers were generally positive, which facilitated their engagement in HIV care" | Study participants included "those who (1) selected male or intersex as their sex labeled at birth, (2) selected woman or trans woman as at least one of their gender identities (n = 54), and (3) had complete data about their access to HIV care (n = 50)." |
| Lacombe-Duncan 2021 [33] | Cross-sectional qualitative study investigating the barriers and facilitators for trans women accessing HIV support, prevention and treatment. Focus groups were conducted with trans women (including women with transfeminine experience) ≥18 years, who also completed a questionnaire. Interviews were conducted with professionals who had provided medical or social services to trans women in the past 2 years. All participants lived in Vancouver, Toronto or Montreal. Focus groups: September–December 2018 Interviews: January– April 2019 | Transgender women, n = 26 Mean age (SD): 40.7 (12.5) Service providers, n = 10 | *Questionnaire findings in trans women:* Ever had an HIV test (n=26): Yes: 88.5% No: 11.5% Results of HIV test (n=23): Positive: 17.4% Negative: 82.6% Have family doctor (n=25): Yes: 88.0% No: 12.0% *Focus group and interview findings:* Barriers to HIV prevention, treatment and support for trans women included: "(a) anticipated and enacted stigma and discrimination in the provision of direct care, (b) lack of provider knowledge of HIV prevention and care needs for trans women, (c) absence of trans-specific services and organisations and (d) cisnormativity in sexual healthcare." Facilitators for HIV prevention, treatment and support for trans women included: "(a) the provision of trans-positive trauma-informed care, (b) autonomy and choice for trans women in selecting sexual health services appropriate to their needs and (c) models for trans-affirming systems change" | "Focus group participants were purposively recruited through the personal and professional networks of the study team with the use of a study flyer distributed via the study team's email listservs and social media accounts." "Potential service providers known to provide care to trans women were recommended by the study team to the first author, and then recruited by email by the first author until 10 interviews were completed." |

*(Continued)*

| Paper | Study design, location, study name and study period | Population Characteristics | STBBI-related findings (95% CI) | Notes |
|---|---|---|---|---|
| Rich 2017 [34] | Analysis of qualitative interviews about sexual HIV risk from transgender men recruited from a longitudinal cohort study (Momentum Health Study) of 719 cisgender and transgender men aged ≥16 years who lived in Metro Vancouver and had had sex with a man in the previous 6 months. Cohort recruitment from February 2012 – February 2014. interviews from November – December 2014 | Transgender GBMSM, n = 11 Median age (IQR): 26 (25–28) Aboriginal: 0% | HIV-negative: 100% "Participants in this study described a number of barriers to accessing healthcare including service provider misperception of their gender-identities and bodies, lack of provider training in trans-competent healthcare and language, and the need to self-advocate and educate as patients." "Participants' narratives suggest that HIV risk for these transgender men is shaped by a diversity of sexual behaviours, including inconsistent condom use, seeking partners online for greater safety and accessing HIV/STI testing and other healthcare services despite facing transition-related barriers." | "All participants who reported female sex assigned at birth identified their current gender identity as a 'Trans-man (F to M)'." |
| Scheim 2017c [35] | Cross-sectional qualitative study (Trans MSM Sexual Health Study) investigating barriers to STI testing among trans men aged ≥ 18 years who lived in Ontario and had had sex with a man in the previous year. Interviews conducted in 2013 | Transgender GBMSM, n = 40 Age 18–24: 25%; 25–34: 48%; 35–44: 23%; 45 +: 5% Aboriginal/ Indigenous: 10% | HIV status: Negative: 85%; Positive: 0%; Don't know: 15% "Participants described a number of barriers to HIV and other STI testing. These included both trans-specific and general difficulties in accessing sexual health services, lack of trans health knowledge among testing providers, limited clinical capacity to meet STI testing needs, and a perceived gap between trans-inclusive policies and their implementation in practice. Two major facilitators were identified: access to trusted and flexible testing providers, and integration of testing with ongoing monitoring for hormone therapy." | "Eligible participants identified as trans men or trans masculine" |
| Scruton 2014 [36] | Cross-sectional online survey (Trans Needs Assessment, Canadian AIDS Society) of trans people in all provinces and the Northwest Territories from September 2013–January 2014 | Transgender people, n = 460 Age < 18: 3%; 18–24: 30%; 25–34: 28%; 35–44: 14%; 45–54: 13%; 55–64: 8%; 65 +: 2%; Prefer not to say: 2% Aboriginal: 6% | Testing for HIV (n = 262): Never tested for HIV: 39% Tested more than one year ago: 35% Tested within the last year: 25% Prefer not to say: 2% Self-reported HIV status: HIV positive: 0.8% Prefer not to say: 1.9% Did not know status: 17.9% Sharing needles (e.g., for drugs, hormones, silicone or saline) in the past 12 months: Never: 99.6% Once: 0.4% 2-6 times: 0% 12 times or more: 0% Sex in the past 12 months: Never: 34% Unprotected sex with a high-risk partner or a partner who does not know their HIV status in the past 12 months: Never: 90.0% Once: 3.9% 2-6 times: 3.5% 12 times or more: 2.7% | Trans people self-identified by answering yes to the question: "Do you now or have you ever considered yourself to be trans* (including transgender, transsexual, intersex, genderqueer or gender nonconforming in any way)?" Not all respondents answered all questions but the sample size for the number of respondents for each question was not given in most cases. |

*(Continued)*

**Table 6.** (Continued)

| Paper | Study design, location, study name and study period | Population Characteristics | STBBI-related findings (95% CI) | Notes |
|---|---|---|---|---|
| Stewart 2022 [37] | Sub-analysis of data relating to transgender, non-binary or two-spirit GBMSM from a larger cross-sectional qualitative study of STI testing experiences of cis and trans gbMSM aged ≥18 years in Ontario who had tested for an STI in the past 12 months. Interviews conducted from June–September 2020 and September–December 2021 | Transgender, non-binary or two-spirit GBMSM, n = 13 Age 18–24: 23%; 25–34: 46%; 35–44: 23% 45–54: 8% Indigenous/First Nations: 23% | "Participants described STBBI testing environments as exclusionary of trans and gender non-conforming people within their procedural policies. Gender segregated clinic hours was named most often as an exclusionary policy that relied on cisnormative expectations that clients identified with one gender, or that clients would automatically know which hours to attend if they were trans." "Often, the onus [was] on the trans client to request proper tests and disclose their trans status because providers assumed their genders based on their appearance or inaccurate documentation" "Participants explained that some service providers lacked the competency to know the sexual risk profiles of [transgender, non-binary or two-spirit GBMSM] and did not provide comprehensive and informed care when testing them for STI." "Participants noted that when documentation, such as medical charts, lab forms, and health cards were incongruent with their gender, this created barriers to testing. [Transgender, non-binary or two-spirit GBMSM] who have documentation that do not reflect their gender are often misgendered in testing settings" | "The term 'trans masculine and non-binary'… refers to participants who were assigned female at birth who may identify as trans men, two-spirit or non-binary, and participants who were assigned male at birth and identified as two-spirit or non-binary." |

AIDS = acquired immunodeficiency syndrome; ART = antiretroviral therapy; cART = combination antiretroviral therapy; CI = confidence interval; cis = cisgender; GBMSM = gay, bisexual and other men who have sex with men; HIV = human immunodeficiency virus; IQR = interquartile range; MSM = men who have sex with men; PrEP = pre-exposure prophylaxis; SD = standard deviation; SMD = standardised mean difference; STBBI = sexually transmitted and blood-borne infection; STI = sexually transmitted infection; VL = viral load; WLWH = women living with HIV.

which analysed qualitative data from a survey of 76 transgender women in five provinces [30], and Lacombe-Duncan 2021 [33].

Participants in Munro 2017 [21] and Everhart 2022 [30] also highlighted integrated services that provide support in navigating the healthcare system as facilitators to access. This was similar to a facilitator to access identified by participants in Scheim 2017c: the integration of STBBI testing with monitoring services for hormone therapy [35].

### Use of healthcare and prevention services (including pre-exposure and post-exposure prophylaxis)

Abramovich 2020 found that the mean number of primary care provider visits in four outpatient clinics in Ontario from January 2012 to December 2016 was 22.55 (standard deviation (SD) 27.15) for the 2,085 transgender people and 13.18 (SD 19.62) for the 10,425 presumed cisgender people, with no statistically significant difference in the number of visits between the groups (p = 0.40) [27]. Arps 2021 found that 74% of 133 transgender and non-binary sex workers had a primary healthcare provider, compared with 81% of 2,236 transgender and non-binary non-sex workers (p = 0.038) [28].

In the Sex Now 2018 survey, published by Rutherford 2021, 76.4% of transgender GBMSM had a regular healthcare provider, compared with 71.7% of cisgender GBMSM (aOR 1.82 [1.30–2.57]). Fewer transgender GBMSM (4.1%) than cisgender GBMSM (12.3%) were taking HIV pre-exposure prophylaxis (PrEP) (aOR 0.56 [0.28–1.00]) while more transgender GBMSM (4.4%) than cisgender GBMSM (2.6%) had ever taken and stopped HIV PrEP (aOR 2.96 [1.45–5.65]).

More transgender GBMSM (38.2%) than cisgender GBMSM (30.6%) were vaccinated for HPV (aOR 1.51 [1.08–2.12]) but vaccination rates against HBV were not statistically different between the groups (58.4% and 67.9% of transgender and cisgender GBMSM, respectively). For all these outcome measures, estimates for non-binary GBMSM in the study were not statistically different from those of cisgender GBMSM [26].

Gilbert 2023 was a cross-sectional online survey of Mpox vaccine uptake among clients of an STI clinic in Vancouver. The survey found that 10/19 (52.6%) of transgender people, 120/174 (69.0%) of cisgender people and 7/13 (53.8%) of people with other gender identifies or who preferred not to say their gender identify received the Mpox vaccine among those eligible [31].

## Findings from expert interviews

All experts agreed that more data are needed on STBBIs in transgender and non-binary people in Canada. A common theme discussed across interviews was the lack of STBBI surveillance data disaggregated by gender identity at the national and provincial and territorial levels. As a result, most prevalence data and other epidemiological information come from research studies focusing on transgender people, studies which were not designed to produce generalisable prevalence estimates and which are highly limited both geographically and in terms of sample size. In several interviews, experts also noted a data gap relating to how STBBI risk is affected by sexual anatomy and gender-affirming surgery and by the intersection of physiology, identity, social determinants of health and sexual behaviour.

One expert suggested that sexual inactivity is relatively common in transgender communities in Canada and that future surveys in these populations should have behavioural entry criteria such as having had sex in the past six months. Some experts thought that existing transgender-specific studies conducted in Canada, such as Trans PULSE Ontario, did not sufficiently capture the subpopulations most at risk of STBBIs, and advocated for future studies focusing on these subpopulations. However, opinions varied on the subpopulations most at risk. Some saw transgender women, transgender GBMSM and transgender and non-binary people who are part of gay sex networks as being at particular risk. Others thought that transgender people with experience of sex work or those with intersecting marginalised identities, such as Black transgender women, were most at risk. Additionally, several experts noted that most studies on transgender and non-binary people in Canada were conducted in large metropolitan areas and advocated for studies focusing on these populations in rural communities, about which little is known. No experts were aware of completed research projects corresponding to our inclusion criteria that we had not already identified through our scoping review search.

## Discussion

It is difficult to quantify the prevalence of any STBBI among transgender and non-binary people in Canada. In this comprehensive scoping review of published data, provincial and territorial reports and the websites of relevant community organisations, we found only one source that provided national-level estimates of the prevalence of any STBBI among the general population of transgender people in Canada: an online survey conducted in 2013–14 in which 0.8% of participants self-reported as living with HIV and 17.9% did not know their status [36]. No sources provided such estimates for non-binary people. The only other study with HIV estimates among the overall transgender population used data collected between 2012 and 2016 in outpatient clinics in three cities in Ontario, finding an HIV prevalence of 1.6%, much higher than in age-matched presumed cisgender people in the general provincial population [27]. Data from both studies should be interpreted with caution, however, as the online survey had an unclear sampling frame and it is hard to know whether patients attending clinics in the three cities were representative of the general population of transgender people in Ontario, or whether transgender PLHIV were particularly likely to seek care at these clinics with expertise working with transgender populations. Either way, these results are not recent, with data collection ending ten and eight years ago, respectively.

More STBBI prevalence data in the peer-reviewed literature are available for subgroups of the transgender population such as transgender women and transgender GBMSM. Estimates of HIV prevalence among transgender women ranged from 1.2% in the Trans PULSE Ontario study (conducted in 2009–10) [19] to 5.8% in the Montreal Toronto Trans Study (conducted in 2018–19) [22]. The higher HIV prevalence in the Montreal Toronto Trans Study is expected as it utilised electronic medical record data from primary care clinics providing HIV care. For transgender GBMSM, national data from the in-person Sex Now 2018 survey found that participants self-reported an HIV prevalence of 5.1%, statistically non-different from the prevalence for cisgender GBMSM, though significantly fewer transgender GBMSM had ever been tested for HIV [26].

Of the studies mentioned so far in this discussion, most were not designed to produce estimates of prevalence that are generalizable to broader transgender populations. However, they are the best available data because most provinces and territories do not publish surveillance data on STBBIs that is disaggregated by gender identity. Quebec has been publishing annual HIV case counts among transgender people since 2002 [15,16], and Ontario began publishing gonorrhoea, chlamydia and syphilis case counts among transgender people in 2019 [12–14]. Beyond these two provinces, we found no published provincial, territorial or national surveillance data for STBBIs in transgender and non-binary people.

Several studies included in this review suggest that a substantial proportion of transgender people are not sexually active. In the Trans PULSE Ontario study, 37% of the general transgender population had not had sex in the past year [17], with sub-analyses of the same dataset showing variation between different population subgroups: 15% of transgender GBMSM [18] and 41% of transgender women [20] had not had sex in the past year. These findings are complemented by the Sex Now 2015 survey, which found that transgender GBMSM had substantially higher odds of not being sexually active than cisgender GBMSM [25]. Again, these data are either geographically limited to a single province or are limited to a subgroup of the transgender population. However, other work in Canada has shown that transgender people are often excluded from the world of dating [38] and one of the experts we interviewed, who worked for a transgender-focused community organisation, suggested that sexual inactivity is common in the communities they interact with. No data were available on the sexual activity of non-binary people and few data are available on injection drug use-related risk in either transgender or non-binary people.

For the testing of STBBIs, studies provided both quantitative and qualitative data. Quantitatively, in Sex Now 2015, transgender GBMSM had significantly lower odds than cisgender GBMSM of having been tested for HIV and other STIs in the past year. However, as transgender GBMSM in the study were significantly less likely to have been sexually active over the same time period, this is probably unsurprising [25]. In Sex Now 2018, statistically similar proportions of transgender and cisgender GBMSM had ever tested for HIV and HCV [26]. No data were available on STBBI testing in the general transgender or non-binary populations. For the treatment of STBBIs, two studies examined cascades of care relating to HIV treatment in small populations of transgender PLHIV [29] and transgender WLWH [32], respectively, with neither group meeting the then-current UNAIDS target of 90% of those receiving ART achieving viral suppression [39].

In four qualitative studies focusing on subgroups of the transgender population in different parts of Canada, participants felt that providers lacked knowledge about the specific sexual health needs of transgender people [33–35,37]. Two studies found that transgender women encountered direct discrimination and stigma when accessing testing and treatment for STBBIs [21,33] while one found that the most commonly cited barrier to ART use among transgender WLWH was concerns about interactions between ART and feminising hormones [32]. Participants in a survey of transgender GBMSM in Ontario called for the integration of STBBI testing with monitoring for masculinising hormone therapy [35] while participants in a study of transgender women in five provinces also saw integrated services as a facilitator to access for STBBI testing and treatment [30]. Two studies of transgender women called for the provision of trauma-informed transgender-specific and transgender-inclusive services to facilitate STBBI testing and treatment [30,33]. No studies examined qualitative experiences of accessing STBBI testing and treatment among non-binary people.

Data on the use of healthcare and prevention services by transgender and non-binary people are limited and varied. The study in outpatient clinics in three cities in Ontario found no statistically significant difference in the number of visits between the transgender and presumed cisgender people [27]. By contrast, the Sex Now 2018 survey found that transgender GBMSM had higher odds of having a regular healthcare provider than cisgender GBMSM [26]. In the same survey, transgender GBMSM were less likely than cisgender GBMSM to be currently taking HIV PrEP but were more likely to have taken it in the past and later stopped. The reasons for stopping PrEP were not explored in the survey. Estimates for non-binary people in the survey (who all identified as GBMSM or as having had sex with another man in the past five years, as well as being non-binary) were not statistically different from those of cisgender GBMSM [26].

We conducted this scoping review and interviewed experts to understand the available data relating to how STBBIs affect transgender and non-binary people in Canada. The overwhelming consensus of the experts was that more data are needed on STBBIs in transgender and non-binary people in Canada, a view backed up by the findings of this scoping review. More robust and inclusive data would help to better guide resources, treatment and prevention efforts and to improve the development of evidence-based, inclusive policy making, in line with the priority of the Federal 2SLGBQTI+ Action Plan mentioned in the Introduction [2].

Several experts noted a data gap relating to how STBBI risk is affected by sexual anatomy and gender-affirming surgery and by the intersection of physiology, identity, social determinants of health and sexual behaviour. We found no data on how gender-affirming surgery affects STBBI risk. Additionally, the studies we included in this review did not provide data disaggregated on intersecting aspects of identity such as race and ethnicity, disability and age that may affect STBBI risk, and did not collect detailed data on sexual behaviour. Collecting more data on these topics in Canada would be valuable.

The findings of this review suggest that different sub-populations of transgender and non-binary people have different levels of risk of contracting STBBIs, in part because they have different levels of sexual activity, and access testing, treatment and prevention services at different rates. However, none of the studies we found collected STBBI prevalence data alongside data relating to behavioural risk exposures and engagement in prevention and care, making it difficult to determine precisely how sexual risk exposures impact STBBI risk in these populations. Bio-behavioural surveillance of STBBIs is conducted with other key populations in Canada and would provide valuable information if conducted with transgender and non-binary people.

Additionally, as Fig 2 illustrates, substantially more research has been conducted with some transgender sub-populations in Canada than others. Most of the papers we found focused on the general transgender population, or on transgender women, transgender GBMSM and non-binary people who also identify as GBMSM. We found no papers focusing on heterosexual transgender men and only one paper focusing on non-binary people who did not specifically identify as GBMSM. The experiences of STBBIs among heterosexual transgender men and non-binary people of a range of sexual orientations would be valuable topics of future study.

To our knowledge, this is the first published scoping review focusing on STBBIs in transgender and non-binary people in Canada. The main strengths of our scoping review are the broad scope of our search, involving both a database search and a substantial grey literature search in English and French, and the broad scope of our inclusion criteria and extracted data, incorporating qualitative, quantitative and mixed-methods data corresponding to five topic areas covering a range of factors affecting STBBI epidemiology and care. The main limitation is the potential for publication bias, as is the case for any evidence synthesis.

In summary, the available evidence on STBBIs in transgender and non-binary people in Canada is variable and scarce. It is not possible to make specific recommendations for public policy based on the evidence as it currently exists. Responding to this knowledge gap would help to better understand the burden of STBBIs and associated determinants of health in these populations in Canada, and to promote health equity in line with the Federal 2SLGBQTI+ Action Plan [2].

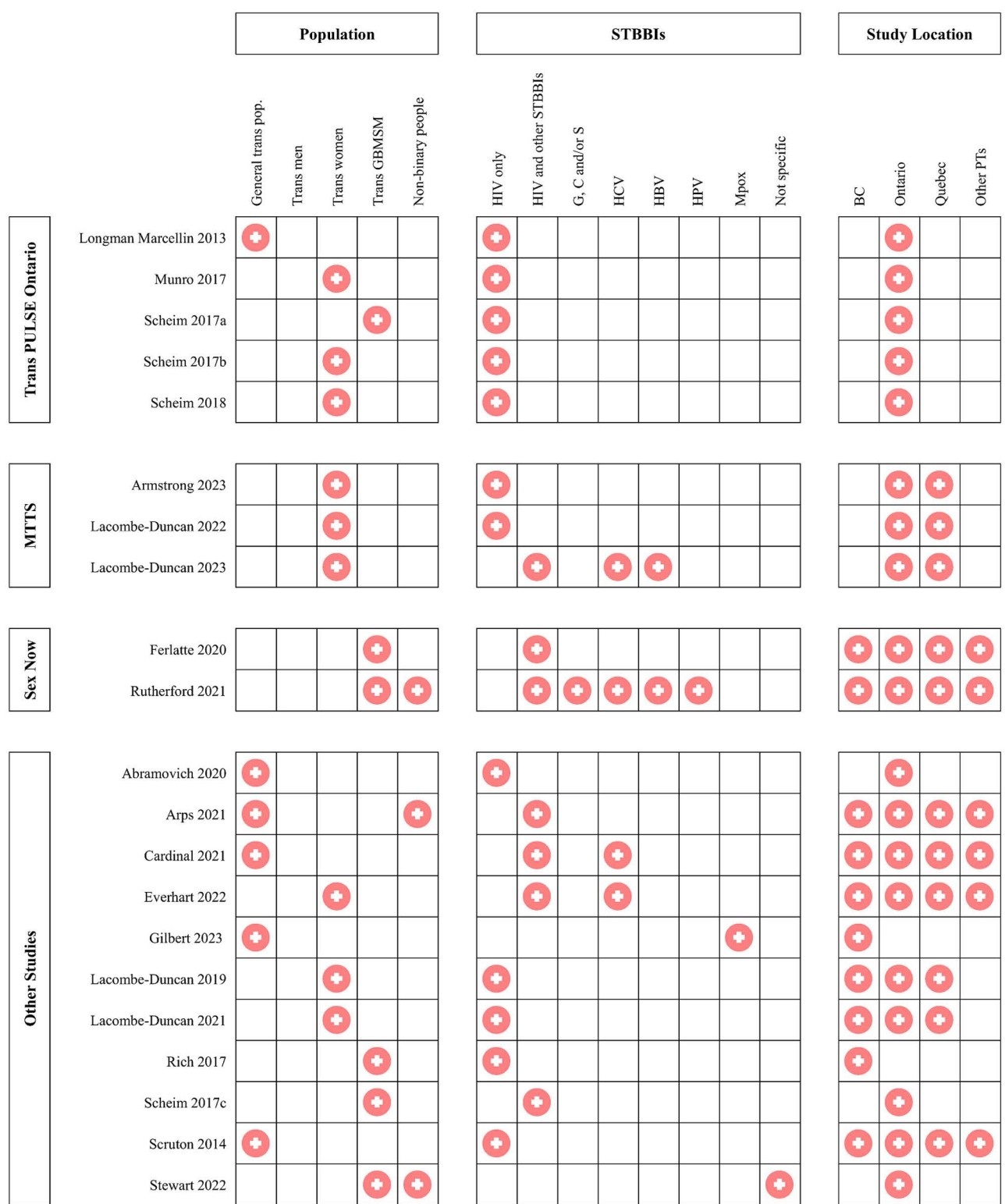

**Fig 2. Population, range of STBBIs reported on and study location for each study.**

## Supporting information

**S1 File. Details of database and grey literature searches.**
(DOCX)

**S2 Table. Reasons for exclusion for studies identified via databases.**
(DOCX)

**S3 Checklist. PRISMA checklist.**
(DOCX)

## Author contributions

**Conceptualization:** Jacob Bigio, Josephine Aho.

**Investigation:** Jacob Bigio, Megan Butler.

**Methodology:** Jacob Bigio, Megan Butler, Swati Sood.

**Project administration:** Jacob Bigio.

**Supervision:** Jacob Bigio, Josephine Aho.

**Validation:** Jacob Bigio, Megan Butler.

**Visualization:** Jacob Bigio, Megan Butler.

**Writing – original draft:** Jacob Bigio.

**Writing – review & editing:** Jacob Bigio, Megan Butler, Swati Sood, Joseph Cox, Beth Jackson, Zack Marshall, Josephine Aho.

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
