## [Decision Letter · Decision Letter 0]

22 Aug 2024

PONE-D-24-30964

Sexually Transmitted and Blood-Borne Infections in Transgender and Non-Binary People in Canada: A Scoping Review

PLOS ONE

Dear Dr. Bigio,

Thank you for submitting your manuscript to PLOS ONE. After careful consideration, we feel that it has merit but does not fully meet PLOS ONE’s publication criteria as it currently stands. Therefore, we invite you to submit a revised version of the manuscript that addresses the points raised during the review process.

We look forward to receiving your revised manuscript.

Kind regards,

Carlos Miguel Rios-González, Ph.D

Academic Editor

PLOS ONE

Journal Requirements:

4. We note that this manuscript is a systematic review or meta-analysis; our author guidelines therefore require that you use PRISMA guidance to help improve reporting quality of this type of study. Please upload copies of the completed PRISMA checklist as Supporting Information with a file name “PRISMA checklist”.

Reviewers' comments:

Reviewer's Responses to Questions

**Comments to the Author**

1. Is the manuscript technically sound, and do the data support the conclusions?

Reviewer #1: Yes

2. Has the statistical analysis been performed appropriately and rigorously? 

Reviewer #1: Yes

3. Have the authors made all data underlying the findings in their manuscript fully available?

Reviewer #1: Yes

4. Is the manuscript presented in an intelligible fashion and written in standard English?

Reviewer #1: Yes

5. Review Comments to the Author

Reviewer #1: Strength of Manuscript: The article addresses a relevant and topical issue on STBBIs (sexually transmitted infections and other reproductive tract infections) in transgender and non-binary people in Canada. The study incorporates a scoping analysis, which was complemented by interviews with 18 experts, including researchers and people working in transgender community organizations. The methodological approach is comprehensive and appropriate for the topic.

Relevance of the Data and Conclusions: The data collected is consistent and supports the conclusions presented. The authors highlight the need for more robust and inclusive data to improve resources, treatment and prevention efforts for these populations, which is in line with the guidelines of the 2SLGBQTI+ Federal Action Plan. In addition, the analysis identifies significant gaps in the current literature, most notably the lack of disaggregated data that considers intersectional aspects such as race, ethnicity, disability and age, which can affect the risk of STBBI.

Recommendations for Improvement: The article could be strengthened with a more detailed analysis of sexual behaviors and their impacts on the health of transgender and non-binary subpopulations. In addition, the inclusion of data on the effects of gender affirmation surgery and other intersections between social identity and health would help to further refine the conclusions. It would also be beneficial if the authors included specific recommendations for public policy and health practices based on the study's findings.

Ethical Considerations and Conflicts of Interest: The article declares no conflicts of interest among the authors, which suggests an unbiased analysis. There is no indication that the study involved the need for ethical approval, presumably because the data was collected secondarily or through non-clinical interviews.

In summary, the manuscript is technically sound and offers significant contributions to the field. With minor revisions and expansions, it could become an essential piece to guide future health policies and practices for transgender and non-binary people in Canada.

Translated with DeepL.com (free version)

6. PLOS authors have the option to publish the peer review history of their article (what does this mean? ). If published, this will include your full peer review and any attached files.

**Do you want your identity to be public for this peer review?** For information about this choice, including consent withdrawal, please see our Privacy Policy .

Reviewer #1: No

---

## [Author Response · Author response to Decision Letter 1]

5 Oct 2024

We thank the reviewer for their helpful comments.

Review Comments to the Author

Reviewer #1: Strength of Manuscript: The article addresses a relevant and topical issue on STBBIs (sexually transmitted infections and other reproductive tract infections) in transgender and non-binary people in Canada. The study incorporates a scoping analysis, which was complemented by interviews with 18 experts, including researchers and people working in transgender community organizations. The methodological approach is comprehensive and appropriate for the topic.

Relevance of the Data and Conclusions: The data collected is consistent and supports the conclusions presented. The authors highlight the need for more robust and inclusive data to improve resources, treatment and prevention efforts for these populations, which is in line with the guidelines of the 2SLGBQTI+ Federal Action Plan. In addition, the analysis identifies significant gaps in the current literature, most notably the lack of disaggregated data that considers intersectional aspects such as race, ethnicity, disability and age, which can affect the risk of STBBI.

Recommendations for Improvement: The article could be strengthened with a more detailed analysis of sexual behaviors and their impacts on the health of transgender and non-binary subpopulations.

We agree that such an analysis would be valuable but due to limitations in the data we found, we cannot comment on how sexual behaviours impact STBBI outcomes. In the discussion, we have added the second sentence below to make this clear: “The findings of this review suggest that different sub-populations of transgender and non-binary people have different levels of risk of contracting STBBIs, in part because they have different levels of sexual activity, and access testing, treatment and prevention services at different rates. However, none of the studies we found collected STBBI prevalence data alongside data relating to behavioural risk exposures and engagement in prevention and care, making it difficult to determine precisely how sexual risk exposures impact STBBI risk in these populations.”

In addition, the inclusion of data on the effects of gender affirmation surgery and other intersections between social identity and health would help to further refine the conclusions.

Unfortunately, we did not find any data on the effects of gender-affirming surgery on STBBI risk. We have added the following sentence to the discussion to make this clear: “We found no data on how gender-affirming surgery affects STBBI risk.” Also, we are unable to comment on the effects of other intersections between social identity and health because, as we had written, “the studies we included in this review did not provide data disaggregated on intersecting aspects of identity such as race and ethnicity, disability and age that may affect STBBI risk.”

It would also be beneficial if the authors included specific recommendations for public policy and health practices based on the study's findings.

We thank the reviewer for the helpful prompt to consider how the evidence synthesised in this scoping review can inform public health. The available evidence is insufficient to make specific recommendations, but this is an important conclusion of our review. We have added the following sentence to the summary paragraph: “It is not possible to make specific recommendations for public policy based on the evidence as it currently exists.” We have added the same sentence to the abstract to highlight its importance and consequently edited the abstract to stay within the word count.

Ethical Considerations and Conflicts of Interest: The article declares no conflicts of interest among the authors, which suggests an unbiased analysis. There is no indication that the study involved the need for ethical approval, presumably because the data was collected secondarily or through non-clinical interviews.

This is correct, we did not require ethical approval for this work.

In summary, the manuscript is technically sound and offers significant contributions to the field. With minor revisions and expansions, it could become an essential piece to guide future health policies and practices for transgender and non-binary people in Canada.

Journal Requirements:

We’ve made changes based on these requirements.

Yes, the submission contains all raw data required to replicate the results of our study. The submission is a scoping review. All included papers or grey literature are publicly available, including all the data we extracted as part of our review, and we have included the full search terms we used in the Supporting Material.

This has been done.

4. We note that this manuscript is a systematic review or meta-analysis; our author guidelines therefore require that you use PRISMA guidance to help improve reporting quality of this type of study. Please upload copies of the completed PRISMA checklist as Supporting Information with a file name “PRISMA checklist”.

This has been done.

We have checked the reference list, which is complete and correct, and we also checked and confirmed that none of the papers have been retracted.

---

## [Decision Letter · Decision Letter 1]

3 Jan 2025

PONE-D-24-30964R1Sexually Transmitted and Blood-Borne Infections in Transgender and Non-Binary People in Canada: A Scoping ReviewPLOS ONE

Dear Dr. Bigio,

Thank you for submitting your manuscript to PLOS ONE. After careful consideration, we feel that it has merit but does not fully meet PLOS ONE’s publication criteria as it currently stands. Therefore, we invite you to submit a revised version of the manuscript that addresses the points raised during the review process.

Many thanks for this revised manuscript which is of high quality and meets publication criteria. I would recommend a further minor revision in line with Reviewer 2's comments to include the interviews of 18 experts in the methodology and results, and if applicable, expanded in the discussion. This would allow the reader to understand how these fit with the data and if further work from this is planned.

We look forward to receiving your revised manuscript.

Kind regards,

Alison May Berner

Academic Editor

PLOS ONE

Journal Requirements:

Reviewers' comments:

Reviewer's Responses to Questions

**Comments to the Author**

1. If the authors have adequately addressed your comments raised in a previous round of review and you feel that this manuscript is now acceptable for publication, you may indicate that here to bypass the “Comments to the Author” section, enter your conflict of interest statement in the “Confidential to Editor” section, and submit your "Accept" recommendation.

Reviewer #1: All comments have been addressed

Reviewer #2: (No Response)

2. Is the manuscript technically sound, and do the data support the conclusions?

Reviewer #1: Yes

Reviewer #2: Yes

3. Has the statistical analysis been performed appropriately and rigorously? 

Reviewer #1: Yes

Reviewer #2: Yes

4. Have the authors made all data underlying the findings in their manuscript fully available?

Reviewer #1: Yes

Reviewer #2: Yes

5. Is the manuscript presented in an intelligible fashion and written in standard English?

Reviewer #1: Yes

Reviewer #2: Yes

6. Review Comments to the Author

Reviewer #1: If the authors have answered all the questions, the manuscript should be considered for publication.

Reviewer #2: This article is incredibly well thought out and strategically planned in order to capture and meet its aims of understanding the available data relating to STBBIs in transgender and non-binary people in Canada. It not only clearly identifies and analyses the data that is available but also importantly identifies gaps in research and publication that can be used to guide future research and data collection that could be used to inform care and sexual health services in a deliberate and meaningful way to serve people with gender expansive identities. Notably there is a clear identification of a scarcity of disaggregated data and therefore, the limitations in application of some the current data. The analysis of the differences in patterns of behaviours of people with different gender identities and sexual orientations along where there are gaps in this data was done with inclusion of the nuances and specifics that can shape this.

The article is insightful and relevant. The scoping analysis is comprehensive and appropriate and demonstrates methodology that is well fitting in exploring this topic where there are such inconsistencies and variations in how data is collected and defined. I commend the authors highly and acknowledge the dedication and quality of this work.

The data collected is put into perspective and linked clearly with the conclusions and discussions and where this fits with future work.

In the discussion section there was a very useful section regarding the interviews with 18 experts which brought important framing to the current climate for transgender and non-binary people in the context of sexual health, however, I wonder why these interviews were not mentioned throughout the article, specifically in the methodology or results and only feature in the discussion as a few sentences. They could be expanded and put into conversation with the data collected in this article or it be indicated how the data from these interviews could be used for additional work on this topic.

Irrespective of this, I think this is a formidable piece of work that could be pivotal in guiding future work for the promotion of improved sexual health outcomes for gender expansive individuals navigating differing intersections of unmet need.

7. PLOS authors have the option to publish the peer review history of their article (what does this mean? ). If published, this will include your full peer review and any attached files.

**Do you want your identity to be public for this peer review?** For information about this choice, including consent withdrawal, please see our Privacy Policy .

Reviewer #1: No

Reviewer #2: No

---

## [Author Response · Author response to Decision Letter 2]

17 Feb 2025

Thank you for your kind comments.

We have followed your advice and included more details of the interviews in both the methods and results sections.

In the methods section, we have written:

“Alongside our scoping review, we also interviewed a convenience sample of 18 experts on how STBBIs affect transgender and non-binary people in Canada. The aim of the interviews was to understand their views on the state of the field and to learn about recent and ongoing research on the topic. Twelve (67%) of the experts were academic researchers, five (28%) worked with transgender-focused community organisations and one (6%) worked in government. Around half of the experts identified as transgender or non-binary.

Interviews were semi-structured. We asked the experts for their views on the state of STBBI research and surveillance among transgender and non-binary people in Canada; about data gaps in the field; about recent and ongoing research in the field; and for suggestions for future work they thought would be valuable. We prepared an interview guide that asked broadly about these topics but discussions were largely guided by the responses of the experts towards the topics they felt were most important in the field. We chose this interview strategy because it mirrored the search strategy of our scoping review, which was deliberately broad and exploratory, aiming to identify data on a range of factors affecting STBBI epidemiology and care in these populations without pre-specifying precise outcomes we were searching for. Interviews were conducted by two authors (JB and MB), in some cases in conjunction with a third author (JA), between December 2023 and April 2024. Interviews lasted between 30 and 60 minutes and were conducted virtually.”

In the results section, we have written:

“Findings from expert interviews

All experts agreed that more data are needed on STBBIs in transgender and non-binary people in Canada. A common theme discussed across interviews was the lack of STBBI surveillance data disaggregated by gender identity at the national and provincial and territorial levels. As a result, most prevalence data and other epidemiological information come from research studies focusing on transgender people, studies which were not designed to produce generalisable prevalence estimates and which are highly limited both geographically and in terms of sample size. In several interviews, experts also noted a data gap relating to how STBBI risk is affected by sexual anatomy and gender-affirming surgery and by the intersection of physiology, identity, social determinants of health and sexual behaviour.

One expert suggested that sexual inactivity is relatively common in transgender communities in Canada and that future surveys in these populations should have behavioural entry criteria such as having had sex in the past six months. Some experts thought that existing transgender-specific studies conducted in Canada, such as Trans PULSE Ontario, did not sufficiently capture the subpopulations most at risk of STBBIs, and advocated for future studies focusing on these subpopulations. However, opinions varied on the subpopulations most at risk. Some saw transgender women, transgender GBMSM and transgender and non-binary people who are part of gay sex networks as being at particular risk. Others thought that transgender people with experience of sex work or those with intersecting marginalised identities, such as Black transgender women, were most at risk. Additionally, several experts noted that most studies on transgender and non-binary people in Canada were conducted in large metropolitan areas and advocated for studies focusing on these populations in rural communities, about which little is known. No experts were aware of completed research projects corresponding to our inclusion criteria that we had not already identified through our scoping review search.”

We have also made some minor changes to the discussion to avoid repetition and expand on some points.

---

## [Decision Letter · Decision Letter 2]

25 Mar 2025

Sexually Transmitted and Blood-Borne Infections in Transgender and Non-Binary People in Canada: A Scoping Review

PONE-D-24-30964R2

Dear Dr. Bigio,

We’re pleased to inform you that your manuscript has been judged scientifically suitable for publication and will be formally accepted for publication once it meets all outstanding technical requirements.

Kind regards,

Daniel Demant, PhD, MPH, GradCertHEd, BAppSocSc

Academic Editor

PLOS ONE

Additional Editor Comments (optional):

Reviewers' comments:

Reviewer's Responses to Questions

**Comments to the Author**

1. If the authors have adequately addressed your comments raised in a previous round of review and you feel that this manuscript is now acceptable for publication, you may indicate that here to bypass the “Comments to the Author” section, enter your conflict of interest statement in the “Confidential to Editor” section, and submit your "Accept" recommendation.

Reviewer #1: All comments have been addressed

2. Is the manuscript technically sound, and do the data support the conclusions?

Reviewer #1: Yes

3. Has the statistical analysis been performed appropriately and rigorously? 

Reviewer #1: Yes

4. Have the authors made all data underlying the findings in their manuscript fully available?

Reviewer #1: Yes

5. Is the manuscript presented in an intelligible fashion and written in standard English?

Reviewer #1: Yes

6. Review Comments to the Author

Reviewer #1: I appreciate the authors for carefully revising the manuscript and incorporating the provided suggestions. The additions to the methods and results sections clarify the conduct and findings of the interviews, strengthening the understanding of gaps in STBBI surveillance among transgender and non-binary people in Canada. Additionally, the revisions in the discussion improve the text’s flow and avoid repetition. I consider that the issues raised in the previous review have been adequately addressed.

7. PLOS authors have the option to publish the peer review history of their article (what does this mean? ). If published, this will include your full peer review and any attached files.

**Do you want your identity to be public for this peer review?** For information about this choice, including consent withdrawal, please see our Privacy Policy .

Reviewer #1: No

---

## [Editor Report · Acceptance letter]

PONE-D-24-30964R2

PLOS ONE

Dear Dr. Bigio,

I'm pleased to inform you that your manuscript has been deemed suitable for publication in PLOS ONE. Congratulations! Your manuscript is now being handed over to our production team.

Kind regards,

on behalf of

Dr. Daniel Demant

Academic Editor

PLOS ONE